

# Leveraging machine learning for the detection of structured interference in Global Navigation Satellite Systems

Imtiaz Nabi[1], Salma Zainab Farooq[1,2], Sunnyaha Saeed[1], Syed Ali Irtaza[2], Khurram Shehzad[3], Mohammad Arif[4], Inayat Khan[5] and Shafiq Ahmad[6]

[1] National Center of GIS and Space Applications (NCGSA), Institute of Space Technology, Islamabad, Pakistan
[2] Department of Electrical Engineering, Institute of Space Technology, Islamabad, Pakistan
[3] Department of Electrical and Computer Engineering, COMSATS University Islamabad, Islamabad, Pakistan
[4] Department of Computer Engineering, Gachon University, Seongnam-si, Republic of South Korea
[5] Department of Computer Science, University of Engineering and Technology, Mardan, Pakistan
[6] Industrial Engineering Department, College of Engineering, King Saud University, Riyadh, Saudi Arabia

Corresponding authors
Mohammad Arif,
mohammadarif911@gachon.ac.kr
Shafiq Ahmad, ashafiq@ksu.edu.sa

## ABSTRACT

Radio frequency interference disrupts services offered by Global Navigation Satellite Systems (GNSS). Spoofing is the transmission of structured interference signals intended to deceive GNSS location and timing services. The identification of spoofing is vital, especially for safety-of-life aviation services, since the receiver is unaware of counterfeit signals. Although numerous spoofing detection and mitigation techniques have been developed, spoofing attacks are becoming more sophisticated, limiting most of these methods. This study explores the application of machine learning techniques for discerning authentic signals from counterfeit ones. The investigation particularly focuses on the secure code estimation and replay (SCER) spoofing attack, one of the most challenging type of spoofing attacks, ds8 scenario of the Texas Spoofing Test Battery (TEXBAT) dataset. The proposed framework uses tracking data from delay lock loop correlators as intrinsic features to train four distinct machine learning (ML) models: logistic regression, support vector machines (SVM) classifier, K-nearest neighbors (KNN), and decision tree. The models are trained employing a random six-fold cross-validation methodology. It can be observed that both logistic regression and SVM can detect spoofing with a mean F1-score of 94%. However, logistic regression provides 165dB gain in terms of time efficiency as compared to SVM and 3 better than decision tree-based classifier. These performance metrics as well as receiver operating characteristic curve analysis make logistic regression the desirable approach for identifying SCER structured interference.

# INTRODUCTION

The market footprint of the Global Navigation Satellite System (GNSS) has recently increased due to its capability of providing passive positioning, navigation, and timing

solutions for a wide variety of applications. GNSS services are deployed as an integral component across multiple domains such as land, marine, and aerial vehicle navigation, Internet of Things, asset tracking, and smart grids. The increased dependence on GNSS raises concern on the safety and reliability of these services (*Chen et al., 2023*; *Gianni et al., 2023*; *Crosara et al., 2023*).

The signals incoming from GNSS satellites are extremely low powered (around −160 dBW), which makes them vulnerable to both unintentional and intentional radio frequency interference (*Borre et al., 2007*; *Arif et al., 2020*). Unintentional interference in the signal can be due to very high frequency (VHF) communication, television, mobile communication, *etc.* (*Thombre et al., 2018*). Intentional interference targets GNSS signals to obtain an illegitimate advantage through jamming and spoofing. Jamming is the transmission of a high-powered signal towards the target device, intending to disrupt communication with the GNSS satellites (*Arif, Kim & Qureshi, 2022*; *Arif & Hasna, 2023*). On the other hand, GNSS spoofing involves structuring and transmission of counterfeit GNSS-like signals to distract, damage, and apprehend GNSS services (*Wang, Kou & Huang, 2023*; *van der Merwe et al., 2023*; *Morton et al., 2021*). Multiple spoofing scenarios can exist depending on the attacker's intention; for instance, the purpose of the spoofer may be to deceive the tracking and monitoring facilities such as fleet management services and commercial drone services (*Khan, Mohsin & Iqbal, 2021*). Spoofing may also be used to gain an unfair advantage in location-based games or access restricted areas without being noticed (*Zhao & Chen, 2017*). Ultimately, the goal of a spoofing attack is taking control of navigation services and deceiving the receiver location.

Spoofing attacks can be divided into three types: simplistic, intermediate, and sophisticated attacks. A simplistic attack employs a GNSS signal simulator for generation of counterfeit GNSS signals. Such attacks can be detected easily due to high signal power and non-synchronization with real signals. Intermediate attacks utilize a receiver to monitor and collect authentic signals and transmit fabricated signals which are synchronized with the real signals and follow the same pattern. The sophisticated attack uses multiple, physically displaced, and synchronized intermediate level spoofers to generate counterfeit signals. The advanced attack was conceptualized to overcome defence based on angle of arrival of received signals; however, it has not manifested itself so far because of the high complexity of involving synchronization and communication processes between each transmitter. As the intermediate attack poses the greatest threat, it has attracted most attention (*Sun et al., 2018a*; *Arif, Wyne & Ahmed, 2020*).

To counteract spoofing, many spoofing detection strategies have been devised. These techniques are based on monitoring signal strength, and time and angle of arrival of signals (*Meng et al., 2022*). Where most of the spoofing detection strategies reported in literature (*Rothmaier et al., 2021*; *Junzhi et al., 2019*; *Psiaki et al., 2014*; *Borio & Gioia, 2016*) are valid for simplistic and intermediate spoofing scenarios, they fail against attacks with careful power and time management. In addition, to avoid spoofing in future GNSS systems, cryptographic Navigation Message Authentication (NMA) similar to GPS military code and the Galileo Open Service Navigation Message Authentication is proposed (*O'Driscoll, 2018*). Although cryptographic security was conceived to be invincible, such signals can

be counterfeited by estimation of the unpredictable symbol while it is being transmitted by the satellite and adding this information into a signal replica generator geared with up-to-date spreading code and carrier replicas forming the security code estimation and replay (SCER) attack (*Humphreys, 2013*; *Gallardo & Yuste, 2020*).

To cater for ever evolving spoofing techniques, this paper proposes machine learning (ML) based spoofing detection. For this purpose, the SCER spoofed signal is used which is the most challenging type of spoofing attacks (*Gallardo & Yuste, 2020*). The contributions of this paper are two-fold, a strategy to classify spoofing signals particularly SCER spoofing is presented and a selection scheme for the best ML classifier for a given situation is devised. Four ML models, logistic regression, support vector machines (SVM), K-nearest neighbors (KNN), and decision tree are trained on three autocorrelation power levels from the code tracking loop in the GNSS receiver. The training data is pre-processed to prevent features with larger magnitudes from dominating the learning algorithms. For training, k-fold cross-validation methodology is employed, wherein the data set is partitioned into six folds to ensure a comprehensive examination of model performance. The classifiers are compared based on accuracy, receiver operating characteristic (ROC) curve analysis, and inference time and the best performing one is selected. Figure 1 represents the normalized time efficiency *vs* the F1-score trade-off; it can be observed that the logistic regression provides the best normalized time efficiency without compromising the performance compared to the SVM (serving as performance baseline) and 50% better than decision trees (serving as the normalized time efficiency baseline).

The rest of this paper has been organized in the following sections. 'Literature Review' provides a detailed literature review and background of the previous work, followed by a mathematical framework of the spoofing menace in 'Mathematical Modeling of GNSS Spoofing'. The methodology, data collection, and experimentation have been discussed in 'Methodology'. The results and their discussion is presented in 'Results and Discussion'. Finally, 'Conclusion' concludes this paper.

## LITERATURE REVIEW

In 2001, the US Department of Transportation published the Volpe report, which emphasized the severity of GNSS interference and spoofing attacks on infrastructures that rely on GNSS services (*Carroll, 2003*). The Volpe report also highlighted the unavailability of appropriate defense mechanisms in off-the-shelf receivers. Additionally, no open-source information was available on the capabilities of commercial-grade spoofers to devise an appropriate counter mechanism. Since then, GNSS spoofing has become a hot topic in the GNSS community. Many researchers have worked to design and develop spoofers to investigate the effect of various spoofing techniques on GNSS receivers.

Typically, a GNSS signal consists of three main parts, including the carrier frequency, ranging code and navigation message. The ranging code provides a basic navigation message encryption layer and helps distinguish the navigation message from multiple satellites in a GNSS constellation. It generally has two versions: a civilian code and a military code. For civilian codes, the structure is openly available in the interface control document for

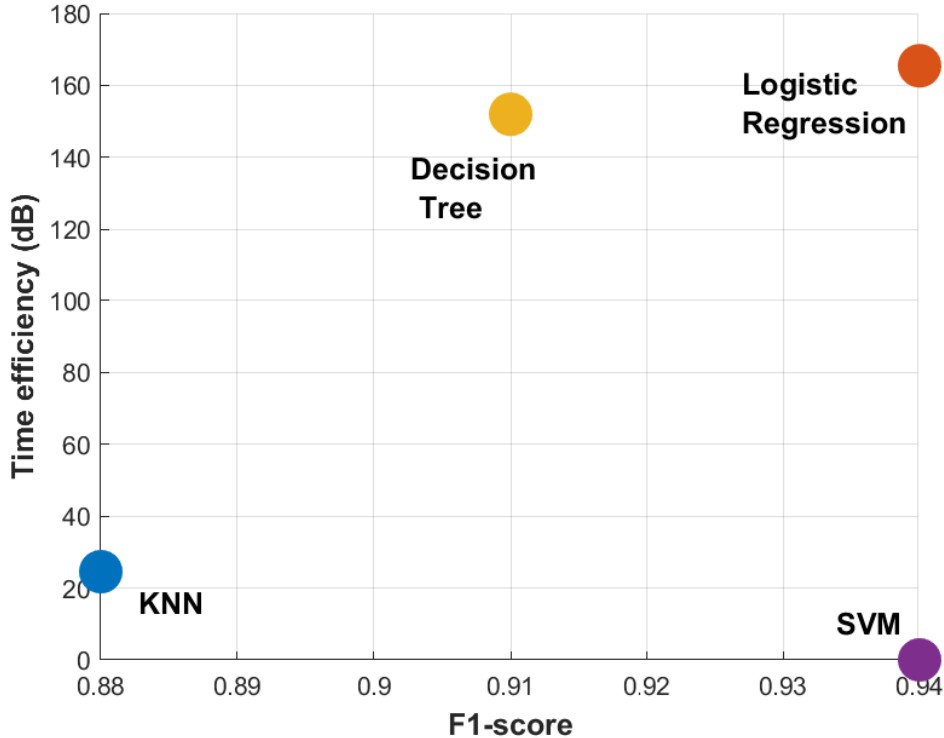

**Figure 1** **Performance comparison on SCER interference ds8 data for time efficiency *vs.* F1-score.** For more detailed analysis refer to 'Results and Discussion'.

each GNSS, making them vulnerable to threats like GNSS spoofing attacks. Military codes have additional built in security features *e.g.*, GPS P(Y) code, an advanced ranging code that flips the encryption key when the anti-spoofing flag is active, making it unpredictable for receivers without the key (*Barker et al., 2000*; *Khan et al., 2022*). Civilian GNSS signals lack signal authentication schemes, with NMA proposed as a solution. Galileo has recently launched its Open Service Navigation Message Authentication (OSNMA) service which allows secure end-to-end transmission from Galileo satellites to OSNMA-enabled GNSS receivers (*Nicola et al., 2022*; *Sarto et al., 2017*). However, other civilian GNSS signals still utilize unsecured communication links and interference signals can easily be modeled and fed to the victim's device using appropriate hardware and software. Furthermore, to breach receivers protected with NMA schemes, the attacker may jam the secured link first and then transmit a spoofing signal in the shadow of the jamming attack to take control of the victim receiver's tracking loop.

GNSS spoofing can be classified into two main categories, meaconing and fabrication attacks (*Lenhart, Spanghero & Papadimitratos, 2021*; *Kerns et al., 2014*). In the case of meaconing attacks, the attacker records a signal at a specific location and time and then replays it at some other location. The signal structure and all signal components remain preserved; thus, it seems like a real signal coming from the satellites (*Hunter, Fillipi & Buesnel, 2020*). In contrast, fabrication attacks involve complete signal generation from a

recorded ephemeris or almanac data set. The code phases and Doppler shift are modeled according to the target spoofing location by calculating the true range, pseudorange, and pseudorange rate between the target location coordinates and the satellite positions computed from the recorded set of ephemeris or almanac data (*Hunter, Fillipi & Buesnel, 2020*; *Warner & Johnston, 2003*; *Larcom & Liu, 2013*). Fabricated spoofing attacks are also referred to as structured interference (*Humphreys et al., 2008*; *Wesson, Rothlisberger & Humphreys, 2012*; *Lo et al., 2009*; *Kuhn, 2004*; *Ledvina et al., 2010*; *Wesson, Rothlisberger & Humphreys, 2011*). Overall based on their characteristics and complexity, spoofing signals can attack the receiver in four configurations (*Dovis, 2015*): (i) Meaconing Attacks (record and replay), (ii) Simplistic Spoofing (attack *via* signal simulator), (iii) Intermediate Spoofing (attack *via* coordinated receiver-spoofers), and (iv) Sophisticated Spoofing (attack *via* coordinated receiver-spoofers with sub-centimeter level information about the victim's position and velocity). The abundance of spoofing mechanisms and multiple attacking scenarios helped researchers and scientists in the GNSS community devise appropriate mechanisms for detecting and mitigating attack scenarios.

A typical GNSS receiver consists of three blocks: radio frequency (RF) front-end, acquisition and tracking block, and navigation data processing block. A right hand circularly polarized L-band antenna receives the signal from GNSS satellites, which is then transferred to the RF front-end which conditions the incoming analog signal and converts it to digital form for onward processing by the acquisition and tracking block. The signal processing in this block follows a channelized structure representing individual satellites. In the acquisition stage, the signal is matched with a local replica of the code and carrier frequency to perform a coarse search for visible satellites' carrier-phase and code-phase. The output of the acquisition stage is forwarded to the tracking stage for fine search, to determine the exact values of carrier-phase (carrier tracking) and code-phase (code tracking).

Spoofing can be detected at the receiving antenna end. In this context, the authors *Montgomery, Humphreys & Ledvina (2009)* propose antenna diversity either through employing multiple receivers or a single-oscillator receiver deploying multiple antennas. After receiving the signal from the same satellite through multiple antennas, a carrier phase measurement difference is formed. If this difference does not fall within the expected phase profiles, based on receiver dynamics, a spoofing attack is declared. Researchers have also studied the use of synthetic aperture antenna arrays with single antenna receivers to detect structured interference (*Nielsen, Broumandan & Lachapelle, 2010*; *Nielsen, Broumandan & Lachapelle, 2011*). However, antenna based countermeasures require modification of the current hardware, which is inefficient and not feasible for off-the-shelf receivers.

Spoofing can also be detected at the signal processing level. In *Jafarnia Jahromi et al. (2012)*, the authors proposed a simple spoofing detection mechanism with the help of carrier-to-noise ratio ($C/N_0$) monitoring. During a spoofing attack, the spoofer introduces an additional noise component from the attacking device front end; this leads to a further drop in authentic signals as they are already weak due to high path losses (*Kaplan & Hegarty, 2006*). Therefore, the receiver can detect an incoming high-powered spoofing signal by comparing the incoming signal *versus* a specifically computed threshold. However, this

detection strategy is only valid for basic spoofing attacks, whereas spoofing signals with careful power management can still deceive the victim device. Instead of monitoring only the $C/N_0$, a better way is to monitor the receiver's automatic gain control (AGC) in combination with the $C/N_0$. The effectiveness of the AGC unit has been verified in *Akos (2012)*, *Manfredini et al. (2018)* in real spoofing environments. In addition, in *Manfredini et al. (2018)*, the relationship between the $C/N_0$ and AGC has been described to differentiate between structured interference and jamming attacks. During jamming attacks, the values of both AGC and $C/N_0$ decrease, whereas in the case of spoofing attacks, the $C/N_0$ value remains relatively constant (depending on the nature of the attack), and the AGC value decreases. While the AGC is highly sensitive to various interference signals, it still is not an effective spoofing detection metric. The AGC value alone cannot provide meaningful insights unless coupled with other receiver information, such as the aforementioned $C/N_0$ strategy. Additionally, the cost of the AGC unit is relatively high, making it unsuitable for mass-market civilian receivers. Another way is to use the signal quality monitoring (SQM) techniques, where statistical binary-hypothesis testing is applied to the correlator outputs from the tracking loops to identify spoofed signals (*Manfredini, Dovis & Motella, 2014*; *Yuan, Li & Lu, 2014*; *Sun et al., 2018b*). However, as spoofing attacks get more sophisticated, SQM based statistical tests fail to produce desired results (*Mahroof et al., 2024*). GNSS spoofing attacks can also be detected by observing the receiver clock offset, but this method is limited to attacks with time spoofing only and fails against methods with synchronized time (*Jovanovic, Botteron & Fariné, 2014*).

With the growing popularity of artificial intelligence and machine learning techniques, scientists and researchers in the GNSS community started to utilize various machine learning and deep learning models for GNSS signal security. For instance, *Feng, Seow & Cao (2022)* used a Gaussian mixture model to differentiate between real and counterfeit signals. During research, it was observed that the trained model could cluster position points generated by a clean signal with 90% accuracy with a single spoofing signal. However, the model's accuracy fluctuates with changes in pseudorange and by increasing the number of spoofed satellite signals. SVM is another classifier used for the detection of deceiving signals as demonstrated by *Zhu et al. (2022)*, *Chen et al. (2022)*. The authors in *Zhu et al. (2022)* use three different spoofing scenarios including SCER spoofing attack and a combination of several features to train the SVM model. They investigate the use of multiple kernel functions and compare them with previous studies. The best SVM kernel resulted in 92.31% accuracy, F1-score of 93.76% and 0.97 area under ROC curve. The authors in *Chen et al. (2022)* used several features to train the SVM model and demonstrated that a larger number of features result in higher performance. However, choosing a large number of metrics to train and test the model renders it impractical for devices with limited processing power and memory. The authors in *Mahroof et al. (2024)* have used KNN for two spoofing scenarios including SCEM attack. The accuracy is seen to be 94.5%. None of the aforementioned research using ML for spoofing detection has used multiple classifiers and compared the performance of different algorithms. In addition, the training parameters are more than three. To test the resilience of Galileo NMA signals, *Gallardo & Yuste (2020)* compared the performance of several classifiers for SCER spoofing attack. It is seen that the detection

accuracy increases if the $C/N_0$ of the input dataset increases. The authors propose ML as an additional GNSS spoofing countermeasure technique, in case the attacker is unable to null the satellite signal.

The summary of this discussion according to the needs of the current situation is as follows:

- GNSS spoofing detection through the $C/N_0$ is ineffective due to a higher false detection rate in multipath scenarios and satellites at low elevation angles. Furthermore, it may completely fail against intermediate and sophisticated spoofing scenarios.
- Some existing techniques require modification of the current hardware equipment and need additional hardware, which is not an economical solution to the current problem. For instance, using multiple antennas may be effective against intermediate-level attacks, but using them on mass-market GNSS devices such as smartphones and tablets may not be feasible.
- The SVM learning strategy is highly effective compared to traditional defense mechanisms; however, it relies on a large number of training features, which requires a large amount of data and eventually more memory and processing power.

Subsequently, this research aims to develop a spoofing detection mechanism based on ML techniques. The method is relatively easy to implement as it uses outputs from the tracking loop of the receiver such that the receiver can be trained to detect the presence of structured interference by using only the code-tracking loop parameters. The novel contributions of this paper are given as follows: (i) the proposed method does not rely on too many features, thus making it suitable for devices with limited computational capabilities, (ii) the proposed method uses code-tracking correlators' power as intrinsic features, making it suitable for scenarios with carefully managed $C/N_0$, and (iii) it is more robust in classifying counterfeit replicas and true signals compared to traditional SQM and $C/N_0$ based detection strategies with significantly better time efficiency as compared to the existing techniques.

## MATHEMATICAL MODELING OF GNSS SPOOFING

The system model of a conventional GNSS spoofing is shown in Fig. 2, where a drone is under attack by a spoofer. The victim receives signals directly from the GNSS satellites and the spoofer simultaneously. To develop a successful spoofing detection strategy, it is important to understand the working mechanism of a spoofing attack and its effect on the victim receiver. The summary of notations used for the mathematical modelling of a signal received under a spoofing attack is provided in Table 1.

Let the true signal incoming from the $k$th satellite be represented by $X^k(t)$, then a generic GNSS signal component, received from the satellite on the L-band can be expressed as (*Psiaki & Humphreys, 2016*):

$$X^k(t) = \sqrt{P_k(t)} C_k\big(t - \tau_k(t)\big) D_k\big(t - \tau_k(t)\big) \cos(2\pi f_L t + \theta_k) \tag{1}$$

where for the $k$th satellite signal $\sqrt{P_k(t)}$ is the signal power, $C_k(t)$ represents the spreading code including subcarrier modulation, $D_k(t)$ is the navigation message, $f_L$ is the L-band

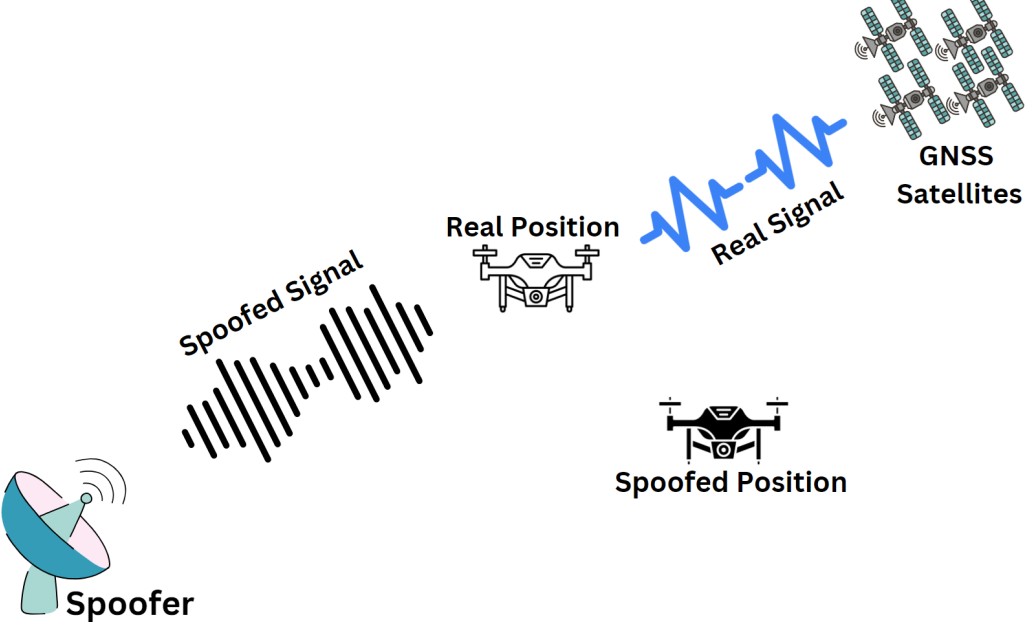

**Figure 2** A conventional spoofing scenario of a drone under attack.

carrier frequency, $\tau_k(t)$ represents the code-phase, and $\theta_k$ is the beat carrier-phase. Message authentication schemes use navigation message as the component to authenticate signals. For a NMA scheme, the received signal is demodulated and the navigation message is decoded for verifying its authenticity. An authentication scheme can be implemented through additional information in the navigation message or assistance from satellite or ground-based augment systems (*Yuan, Tang & Ou, 2023*).

Assuming that atmospheric errors can be corrected by the receiver, the real received signal $X_r(t)$ from total $N_{los}$ number of line of sight satellites also contains residual errors. The signal can be expressed as (*Psiaki & Humphreys, 2016*):

$$X_r(t) = \sum_{k=1}^{N_{los}} X^k(t) + \zeta(t) \tag{2}$$

where $\zeta(t)$ represents noise in real signals due to signal errors such as multipath delay, thermal noise, and receiver noise.

The counterfeit signal $\hat{X}^k(t)$ for $k$th satellite can be modeled as follows:

$$\hat{X}^k(t) = \sqrt{P_{ks}(t)} C_k(t - \tau_{ks}(t)) \hat{D}_k(t - \tau_{ks}(t)) \cos(2\pi f_L t + \theta_{ks}) \tag{3}$$

For Eq. (3), to deceive the receiver, each spoofed signal must have the same spreading code $C_k(t)$ as the corresponding actual signal. The parameter $\hat{D}_k(t)$ is the spoofer's best estimate of the navigation signal's bit stream. The amplitude $\sqrt{P_{ks}}$, code-phase $\tau_{ks}$ and carrier phase $\theta_{ks}$ of the spoofed signal will differ from their true counterparts according to the type of attack (*Psiaki & Humphreys, 2016*).

**Table 1  Brief description of the notations.**

| Notation | Description |
|---|---|
| $k$ | $k$th satellite |
| $(.)_s$ | Spoofed signal |
| $t$ | Receiver time |
| $X^k(t)$ | Signal received from $k$th satellite |
| $P_k(t)$ | Received signal power |
| $C_k(t)$ | Ranging/spreading code |
| $D_k(t)$ | Navigation message |
| $\tau_k(t)$ | Code-phase |
| $\theta_k(t)$ | Beat carrier-phase |
| $X_r$ | Real received signal |
| $\zeta(t)$ | Noise in real signal |
| $\hat{X}^k$ | A counterfeit signal |
| $X_c(t)$ | Sum of all counterfeit signals |
| $\hat{D}_k(t)$ | Navigation signal estimated by spoofer |
| $X_m$ | Sum of multipath signals |
| $Y_s(t)$ | Total signal received under a spoofing attack |
| $\eta(t)$ | Total noise in received signal |
| $N_c$ | Number of counterfeit signals |
| $N_{los}$ | Number of line-of-sight satellite signals |
| $n_s(t)$ | Noise figure contributed by spoofer |
| $f_L$ | $L$-band carrier frequency |
| $C/N_0$ | Carrier-to-noise ratio |
| $T$ | Time efficiency in $dBs$ |
| $I$ | Data from real branch |
| $Q$ | Data from quadrature branch |
| $P$ | Probability of event occurrence |
| $d(x,y)$ | Euclidean distance |
| $TP$ | True positive |
| $TN$ | True negative |
| $FP$ | False positive |
| $FN$ | False negative |

The spoofing attack $X_c(t)$ can be written as:

$$X_c(t) = \sum_{l=1}^{N_c} \hat{X}^l(t) + n_s(t) \tag{4}$$

In Eq. (4), $\hat{X}_l(t)$ is the $l$th counterfeit signal from a sum of $N_c$ number of counterfeit signals where in most cases $N_c \leq N_{los}$, and $n_s(t)$ represents the additional noise figure contributed by the spoofing front-end. These counterfeit replicas generated by the attacker are affected by inconsistencies concerning the real signal due to possible demodulation and estimation errors, along with imperfect synchronization to the real signal.

The received signal under a spoofing attack $Y_s(t)$ is a sum of the real received signal $X_r(t)$, the counterfeit signal $X_c(t)$, and sum of multipath signals $X_m(t)$.

$$Y_s(t) = X_r(t) + X_c(t) + X_m(t) + \eta(t) \tag{5}$$

where $\eta(t)$ represents the total signal noise, which is assumed to be additive white and Gaussian in nature. The impact of multipath on the receiver under a spoofing attack is beyond the scope of this research, therefore, it will not be discussed here. Noise is also ignored as it does not have much effect on position in the present scenario. Therefore for spoofing detection , it is assumed that the receiver receives only true and counterfeit signals.

For NMA schemes, the unpredictable part of the signal lies only in the low-rate data bis $D_k(t)$ allowing vulnerability to SCER attacks (*Psiaki & Humphreys, 2016*). When constructing an SCER attack, the attacker first receives and tracks the authentic GNSS signals considering each symbol unpredictable. It then estimates the unpredictable symbol and synthesizes a fake signal after some delay. Based on delay, there are two different types of SCER attacks (*Humphreys, 2013*; *Gallardo & Yuste, 2020*)

- Zero-latency: The delay of the spoofed signal is considered to be zero at the beginning of the attack. The delay is gradually increased, avoiding effects easily noticeable in the tracking loops of the victim.
- Non-zero-latency: Non-zero delay is present in the spoofer-generated signal. If the delay is significant, to avoid tracking jumps in victim's receiver, at the beginning of the attack, the spoofer may jam the authentic signals for some time interval, thus widening the victim's timing uncertainty.

## METHODOLOGY

The proposed technique is shown in Fig. 3. The methodology is segmented in three sections: data selection, test metrics, and ML based spoofing signal detection. Data processing in each section prepares it for next stage. This includes reading and converting binary data files to extract signal features using open source GNSS-Software Defined Receiver (GNSS-SDR) (*Fernandez-Prades et al., 2011*). For the classifier selection, the selected test metrics are scaled and adjusted at zero mean and unit variance to ensure that all features contribute equally to training the model.

### Data selection

To test the efficacy of ML techniques for antispoofing, Texas Spoofing Test Battery (TEXBAT), an open source data set provided by the Radio Navigation Laboratory of the University of Texas, USA was used (*Humphreys et al., 2012*). The dataset consists of clean static and dynamic signals along with eight carefully created spoofing signals under specific scenarios using the respective clean signals. The procedure through which the signals are recorded and spoofing signal generated is given in *Humphreys et al. (2012)*. The test battery was extended as Oak Ridge Spoofing and interference test battery for scenarios 1–6 (*Albright et al., 2020*). The successful detection to all spoofing attacks in TEXBAT can

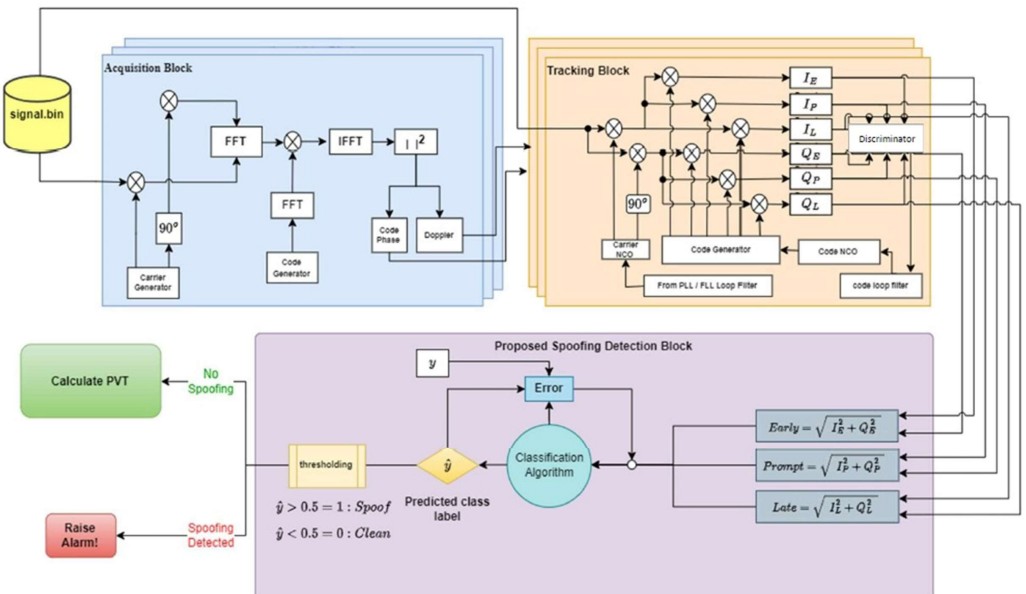

**Figure 3** Proposed spoofing detection strategy.

**Table 2** Description of TEXBAT clean static and ds8 signals used in the paper.

|                   | Clean static      | ds 8              |
| ----------------- | ----------------- | ----------------- |
| **Duration**      | 0–420 s           | 0–420 s           |
| **Valid length**  | 420 s             | 310 s             |
| **Sample size**   | 155,000           | 210,000           |
| **Samples format**| I/Q interleaved   | I/Q interleaved   |
| **Spoofing type** | N/A               | time push         |
| **Spoofing injected** | N/A           | 110 s             |
| **Pull off start**| N/A               | 136 s             |

serve to certify that a civil GPS receiver is spoof resistant (*Humphreys, 2012*). Since we are analyzing the NMA attack type situation, we will be using the eighth spoofing scenario termed as ds8. It is a subtle power-matched time push scenario such that the spoofer attempts to approximately match its ensemble power to that of the authentic signals. The spoofer attempts carrier phase alignment between the spoofing and authentic signals by treating every received navigation data bit as unpredictable, guessing the value of the data bit in real time, thus forming the zero-delay SCER attack. Approximately 50 μs after each navigation data bit transition boundary, the spoofer's guess of the data bit value is correct to a very high probability (*Humphreys, 2016*). Description of both clean static and ds8 signals can be seen in Table 2.

The files are downloaded as binary files and processed through GNSS-SDR. The receiver was configured to a sampling frequency of 25 MHz and centered to a carrier frequency

of GPS $L_1$, and the navigation output was set to 500 Hz. Finally, the tracking loop dump feature was set to true, providing each channel's data in dedicated files.

## Test metrics

A delay locked loop (DLL) is employed for tracking the spreading code of each satellite in a different channel. The DLL utilizes code discriminators as a feedback mechanism to detect delays between the incoming signal and the locally generated replica. For this time-advanced early (E) and time-delayed late (L) versions of the locally generated prompt (P) signals are deployed. Usually, half-chip spacing is maintained between the DLL's early and late correlator arms. Generally, software-based GNSS receivers utilize the normalized early minus late power discriminator, which is given as in Eq. (6) (*Borre, López-Salcedo & Bhuiyan, 2022*).

$$\epsilon_{NNMEL}^{DLL} = \frac{1}{2} \frac{\sqrt{I_E^2[k]+Q_E^2[k]} - \sqrt{I_L^2[k]+Q_L^2[k]}}{\sqrt{I_E^2[k]+Q_E^2[k]} + \sqrt{I_L^2[k]+Q_L^2[k]}} = \frac{1}{2} \frac{abs(E[k])-abs(L[k])}{abs(E[k])+abs(L[k])} \qquad (6)$$

where $\epsilon_{NNMEL}^{DLL}$ is the discriminator function. For the DLL, $I_E$ and $I_L$ represent early and late correlation outputs for the in-phase (I) branch, $Q_E$ and $Q_L$ represent early and late correlation outputs for the quadrature (Q) branch, and $abs(E[k])$ and $abs(L[k])$ represent the early and late correlation power levels for the $k$th sample. Successful tracking yields a decoded navigation message for calculating the user position, velocity, and time.

All this processing is done by GNSS-SDR on the input binary file, where several parameters such as $C/N_0$, absolute early, late and prompt power, carrier Doppler, *etc.*, are extracted for each channel. The output of the six DLL correlators from the I and Q arm are employed to determine the absolute early, late, and prompt correlation power by GNSS-SDR as follows:

$$abs(E) = \sqrt{I_E^2 + Q_E^2} \qquad (7)$$

$$abs(P) = \sqrt{I_P^2 + Q_P^2} \qquad (8)$$

$$abs(L) = \sqrt{I_L^2 + Q_L^2} \qquad (9)$$

The proposed spoofing detection and classification framework uses $abs(E)$, $abs(P)$, $abs(L)$ as test metrics for training the machine learning model in stage 3 as is used as shown in Fig. 3.

## Machine learning based spoofing signal detection

Supervised ML is used to train on spoofed signals to make predictions. Since there are only two cases, spoofed and real signals, logistic regression and binomial classification models are employed. The models are trained, tested, and validated to devise an accurate spoofing detection mechanism so as to pick one with higher accuracy of spoofing detection with

lower computational cost. For training the model, clean epochs are assigned label '0' and spoofed epochs are assigned label '1'. Four most common and favored binary ML models are used: (i) logistic regression, (ii) K-nearest neighbors (KNN) classifier, (iii) decision tree classifier, and (iv) support vector machines (SVM) classifier. An appropriate classifier selection requires centering each feature of the training data to zero mean and unit variance and normalizing it to bring all the features to a uniform scale to ensure regularity of the data for training the model. These pre-processing steps help to prevent features with larger magnitudes from dominating the learning algorithm.

### *Logistic regression*
The Logistic regression model is one of the simplest models that distinguishes the given input features into binary classes by applying a nonlinear log transformation sigmoid function to the odds ratio as given in Eq. (10):

$$odds = \frac{P}{1-P} \rightarrow logit(P) = ln\left(\frac{P}{1-P}\right) \tag{10}$$

where $P$ is probability of event occurrence. If the linear regression is given as:

$$y = \omega_0 + \omega_1 X_1 + \cdots + \omega_n X_n \tag{11}$$

here $X_i (i \in 1, \ldots, n)$ represents the underlying variables explaining the model behavior and $\omega_i$ represents the contribution of each variable $X_i$, respectively. Then the logistic regression can be applied as follows:

$$\frac{P}{1-P} = e^y \rightarrow P = \frac{e^y}{1+e^y} \rightarrow P = \frac{1}{1+e^{-y}} \tag{12}$$

### *K-nearest neighbors*
The KNN algorithm is a non-parametric classification technique that does not make any presumption on the elementary data. It is an effective machine learning model widely used for classification problems that take labeled training data broken down into different classes to predict unlabeled data. KNN performs classification of the data based on the nearest neighbors using Euclidean distance. For instance, given $x_i$ where $i \in 1, 2, 3, \ldots, n$ is an input feature class to be given to the model and $y_i$ is the target label, then the model calculates the Euclidean distance $d(x, y)$ as follows:

$$d(x, y) = \sqrt{\sum_{i=1}^{n}(x_i - y_i)^2} \tag{13}$$

### *Decision tree*
The decision tree classifier builds a tree structure that contains internal nodes for tests on attributes and branches for the test outcome. The model uses leaf nodes for holding class labels and works recursively by splitting the data into subsets until a stopping criterion is satisfied. The data is split based on the Gini impurity, which measures the randomness of the subsets. The information for class $D$ is calculated as follows:

- First, calculate the entropy associated with the class $D$

$$\text{info}(D) = -\sum_{i=1}^{m} p_i log_2 p_i \tag{14}$$

In Eq. (14) the term $p_i$ is the probability that given instance belongs to the class D.

- Calculate the entropy after splitting D to a new branch $D_i$ where $D_i \subset D$

$$\text{info}_a(D) = \sum_{i=1}^{V} \frac{|D_i|}{|D|} \times \text{info}(D_i) \tag{15}$$

- Calculate the information gain associated with new branch

$$\text{Gain}(A) = \text{info}(D) - \text{info}_a(D) \tag{16}$$

In Eqs. (15) & (16), info(D) is the average amount of information required by the system to identify the given class label. Similarly, the term $\frac{|D_j|}{|D|}$ is the weight of the $j$th partition, and $\text{info}_a(D)$ is the expected information for the classification of the tuple D. A new branch is created if the information gain is greater than a certain threshold according to Eq. (16), where $A$ is the highest information gain and is chosen as the splitting attribute at the nodes.

### *Support vector machine*

SVM processes the input feature vector containing $n \times 3$ parameters of the DLL discriminator output and produces the target label vector $y_i$ of dimension $n \times 1$ containing either 1 or 0 (indicating spoofing and clean signals, respectively) for $n$ epochs. SVM classifies the input feature vector as 1 or 0 through a hyperplane defined as follows:

$$\omega^T \mathbf{x} + \mathbf{b} = \mathbf{0} \tag{17}$$

Here $\omega^T$ is the normal hyperplane vector, $\mathbf{x}$ is the samples vector, and $b$ is the bias constant. Given the DLL discriminator parameters $\mathbf{z}$, the support vector classifier performs a binary classification as in Eq. (18).

$$y_i = \begin{cases} 0, (\omega^T \mathbf{z} + b) \leq -1 \\ 1, (\omega^T \mathbf{z} + b) \geq +1 \end{cases} \tag{18}$$

The optimal hyperplane for the classification can be obtained as follows (*Zhu et al., 2022*):

$$\max_{\omega, b} \min_{x} \{ \|\mathbf{x} - \mathbf{z}\| : \mathbf{x}, \mathbf{z} \in \mathbb{R}^N, (\omega^T \mathbf{x} + b = 0) \}. \tag{19}$$

## RESULTS AND DISCUSSION

### Choice of training data

This section presents the motivation for choosing the DLL output signal power as test metric. For ds8, the spoofing signal is injected between time interval 110–130 s. (*Humphreys, 2016*).

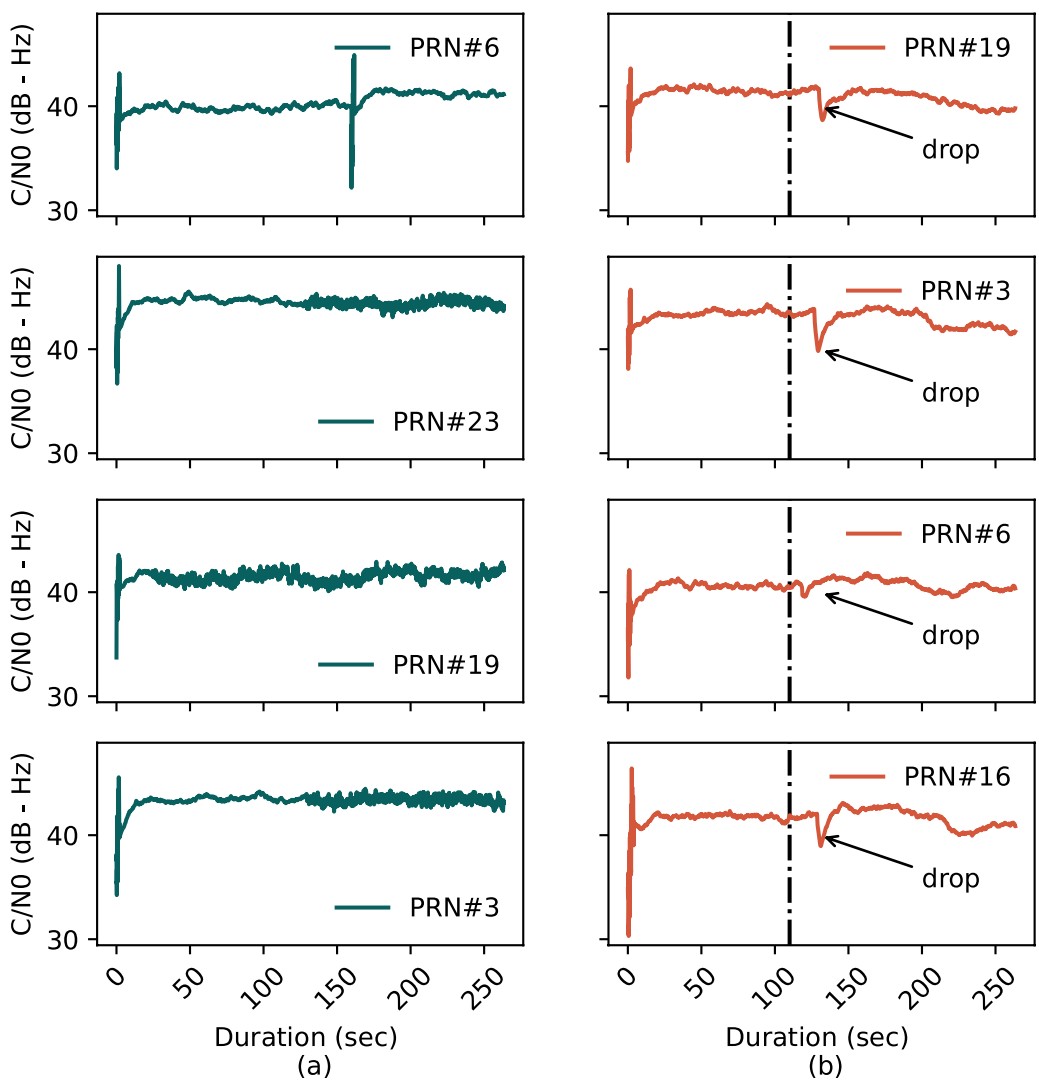

**Figure 4** $C/N_0$ plots of the first four channels of (A) clean static data and (B) spoofing (ds8) data.

### $C/N_0$ Magnitude

The $C/N_0$ values for each channel in ds8 file are one of the features extracted by GNSS-SDR. Figures 4A and 4B show the $C/N_0$ of the first four satellites from clean and spoofing data sets, respectively, where the black dashed line represents the instant where the spoofing signal is injected. The figure shows that the $C/N_0$ of the clean static and ds8 signals are almost identical except for a small kink or drop in the ds8 signal on epochs where the tracking channel locks on the spoofing signals. The gradual blending and subsequent locking of spoofing signals manifests that ds8 is an advanced scenario having carefully managed power, and using a basic $C/N_0$ based detection metric may not be effective against this signal.

***DLL early, late, prompt power levels***

Figure 5 shows the plots for early, prompt, and late correlation power outputs for the same satellites as in Fig. 4 with Fig. 5A representing clean static and Fig. 5B representing the ds8 signal. Even though the $C/N_0$ of the spoofing signal is carefully managed for the ds8 scenario, there is still a need to thaw the real signals in order for the receiver to track counterfeit signals. For this purpose, the power of the spoofing signal must be slightly higher than real signals for the victim receiver to lock it. As shown in Fig. 4, the signal's power is carefully managed and does not display higher than average levels at any time. However, in Fig. 5B, it is seen that there is an abrupt correlation power change at the instant the spoofing signal is injected in the real signal. It can therefore be concluded that the receiver code tracking loop is sensitive to even slight variations in signal power and for this reason DLL correlators' power magnitudes are chosen to train the model.

## Comparative analysis between ML models

Various metrics can quantitatively evaluate different aspects of a classification algorithm. A positive instance or 1 indicates spoofed signal and negative instance or 0 indicates real signals. A true positive (TP) indicates that a the value under consideration is correctly predicted as spoofed signal. A true negative (TN) represents a sample correctly identified as non spoofing. On the other hand, a false positive (FP) and false negative (FN) refer to wrong predictions. ML models are assessed against several parameters: accuracy, precision, recall (sensitivity), Specificity and F1-score (*Colliot, 2023*).

Accuracy is the measure of a model's correct prediction with respect to all the predictions that the model has made. It is defined as:

$$Accuracy = \frac{TP + TN}{TP + FN + FP + TN} \tag{20}$$

Precision is the measure of correctly predicted positive values out of all positive values predicted.

$$Precision = \frac{TP}{FP + TP} \tag{21}$$

Recall or sensitivity is the measure of correctly predicted positive values out of total present positive values.

$$Recall = \frac{TP}{TP + FN} \tag{22}$$

Specificity is the fraction of negative values actually classified as negative.

$$Specificity = \frac{TN}{TN + FP} \tag{23}$$

The F1-score balances precision and recall for binary classification models. It is defined as the harmonic mean of precision and recall.

$$F1 - score = \frac{2 \times Precision \times Recall}{Precision + Recall}. \tag{24}$$

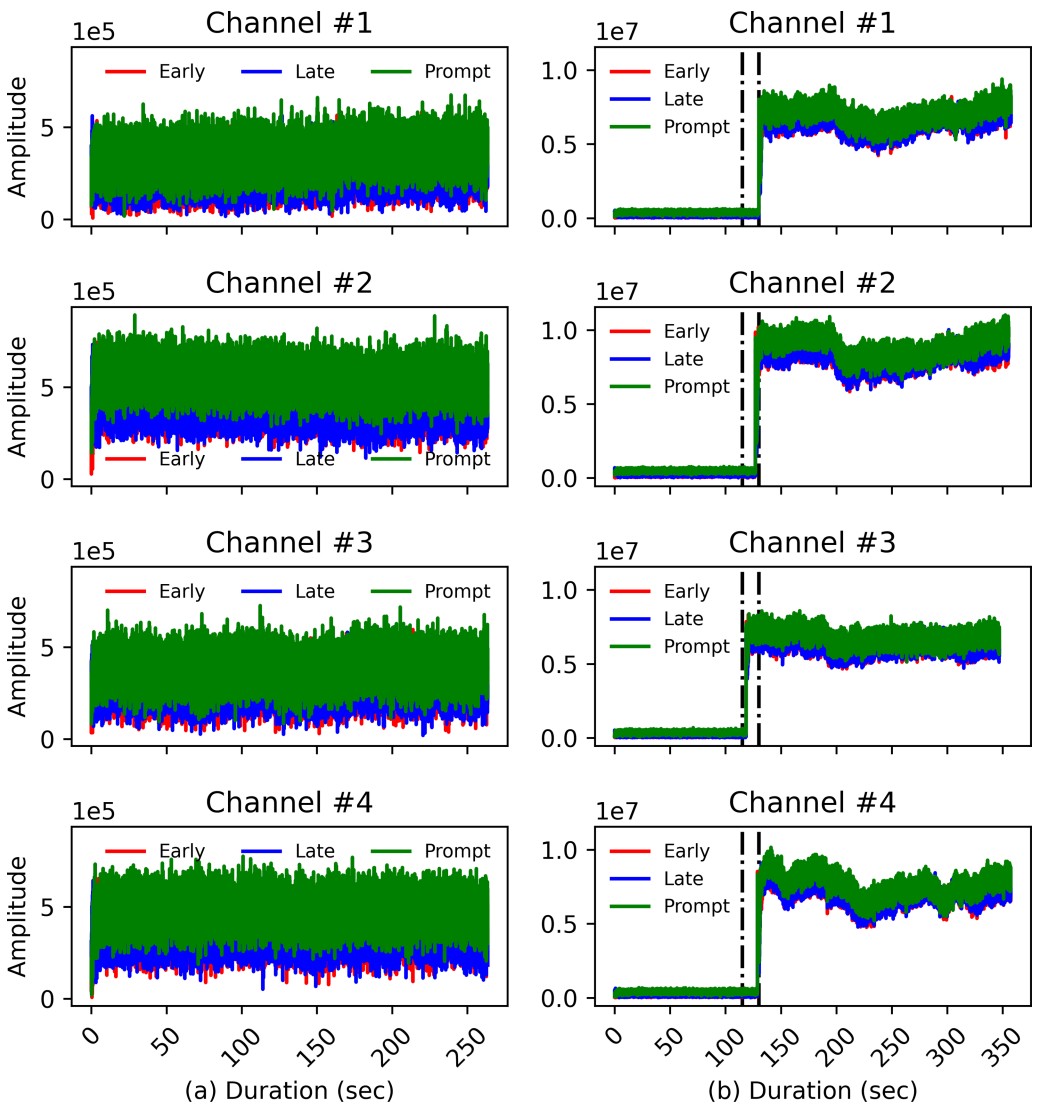

**Figure 5** DLL response for the (A) clean static signal and (B) spoofing injected signal.

### Accuracy score

For testing the accuracy of models, 80% of data from the first channel was selected for training, while 20% was used for testing and validation. K-fold cross-validation methodology is employed, wherein the data set is partitioned into k-folds to ensure a comprehensive examination of model performance. The data set is also shuffled k-times to introduce an element of randomness that guards against unintentional bias. For this research, $k = 6$ is chosen to allow thorough scrutiny and preservation of a reasonable sample size for each evaluation. Cross-validation score for each iteration gauges ML model's accuracy by assessing its performance on the scaled training data. The achieved accuracy for all folds for each ML model can be seen in the box and whisker plot of Fig. 6. It can be observed that logistic regression and SVM have maximum accuracy, followed

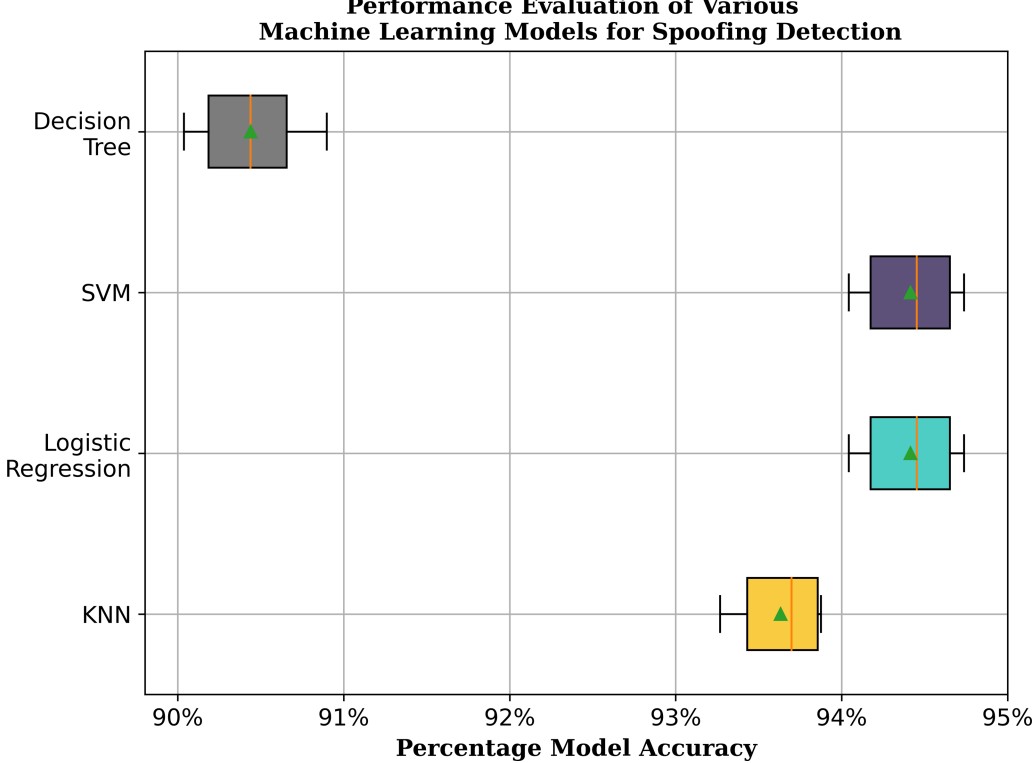

**Figure 6** Comparison of cross validated accuracy of the chosen ML models for spoofing and real signal classification.

by the KNN classifier, while the decision tree model has the lowest accuracy. The average accuracy of these models is also the same.

### Precision, Recall, F1-score

The precision, recall, and F1-score for all four machine learning models rounded to two decimal places can be seen in Table 3. As defined earlier, precision quantifies the accuracy of positive predictions, while recall measures the model's ability to identify all positive instances. For all ML models, precision and recall are high for both real and spoofed signals indicating that the data is not imbalanced. F1-score can provide a single value to gauge the overall performance with reference to precision and recall (*Colliot, 2023*). Macro average, suitable for balanced data, computes the F1-score for each class independently and then takes the average, whereas weighed average, more appropriate for unbalanced data, assigns more weight to classes with greater samples. It is seen that for logistic regression and SVM macro average F1-score is 94% whereas for KNN and decision tree, it is 91% and 88% respectively proving the superiority of these models w.r.t F1-score. Moreover, beside the F1-scores, the proposed model provided a higher recall, which is important for the GNSS spoofing scenario to counter even most intricate of the spoofing attack.

**Table 3 Performance parameters of all four classification models.**

| Model | | Precision | Recall | F1-Score |
|---|---|---|---|---|
| | real | 0.84 | 1.00 | 0.92 |
| | spoof | 1.00 | 0.92 | 0.96 |
| | accuracy | | | 0.94 |
| | macro avg | 0.92 | 0.96 | 0.94 |
| **Logistic regression** | weighted avg | 0.95 | 0.94 | 0.94 |
| | real | 0.84 | 1.00 | 0.92 |
| | spoof | 1.00 | 0.92 | 0.96 |
| | accuracy | | | 0.94 |
| | macro avg | 0.92 | 0.96 | 0.94 |
| **Support vector machines classifier** | weighted avg | 0.95 | 0.94 | 0.94 |
| | real | 0.84 | 0.93 | 0.89 |
| | spoof | 0.97 | 0.92 | 0.94 |
| | accuracy | | | 0.92 |
| | macro avg | 0.91 | 0.93 | 0.91 |
| **K Neighbors classifier** | weighted avg | 0.93 | 0.92 | 0.93 |
| | real | 0.84 | 0.83 | 0.84 |
| | spoof | 0.92 | 0.93 | 0.93 |
| | accuracy | | | 0.90 |
| | macro avg | 0.88 | 0.88 | 0.88 |
| **Decision tree classifier** | weighted avg | 0.90 | 0.90 | 0.90 |

## *Area under the ROC curve*

Area under the curve of ROC is plotted for all the four models as seen in Fig. 7 with specificity plotted along the $x$-axis and sensitivity plotted along the $y$-axis. The ROC curve is plotted across all possible thresholds and area under this curve quantifies the model's ability to distinguish between positive and negative classes. A higher area indicates better discrimination and, therefore, better model performance. If the ROC curve is close to the upper left corner (coordinate (0,1)), the model has high sensitivity and specificity, achieving a good balance between true positives and true negatives. A curve that lies closer to the diagonal line suggests poor discrimination. The further the curve is from the diagonal, the better the model's performance. The TP rate in this situation is near 1, proving the trained model's high accuracy. Specifically, the FP rate is greater than TP rate, enabling the model to discriminate the signals suspected as counterfeit which is plausible given the severity of misclassifying the spoofed signal as the original signal. Moreover, it can be observed from Fig. 7, that area under ROC curve for both logistic regression and SVM is same and greater than the other two models.

## *Computation time for ML models*

Finally, inference time is calculated for each ML model. It is found that the logistic regression took the least time during the classification, as seen in the Table 4, thus logistic regression model is finally selected as the optimum for this experiment. The processing speed difference between logistic regression and SVM can be attributed to the former's

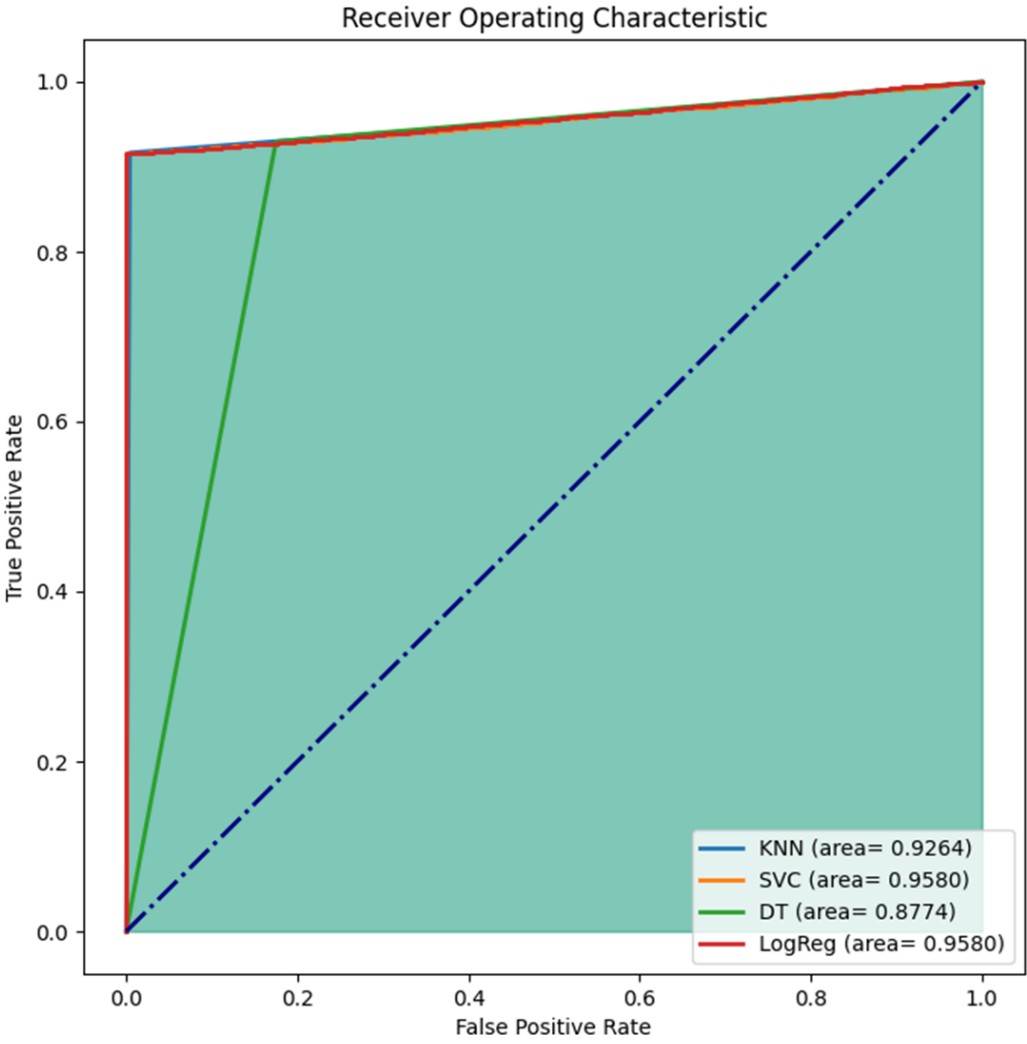

**Figure 7** **Area under the curve analysis for ROC curve.** The true positive rate (sensitivity) is plotted along $y$-axis. A high true positive rate indicates that the model effectively identifies positive instances. The false positive rate (specificity) is plotted on the $x$-axis. A low false positive rate indicates that the model does not classify negative instances as positives.

linearity allowing faster predictions compared to the complexity of computing kernel functions for the latter. The time efficiency $T$ is calculated while considering SVM as the baseline and then converted to $dBs$ using Eq. (25).

$$T_i = 20log_{10}\frac{InferenceTime_{SVM}(sec)}{InferenceTime_i(sec)} \quad i \in \{\text{KNN, Logistic Regression, DT, SVM}\}. \quad (25)$$

## Validation of proposed ML based classifier

In this section the performance of the proposed logistic regression classifier is validated using clean and spoofed signal from the TEXBAT database. First, the clean static data was provided to the trained logistic regression model. In Fig. 8, it can be observed that the

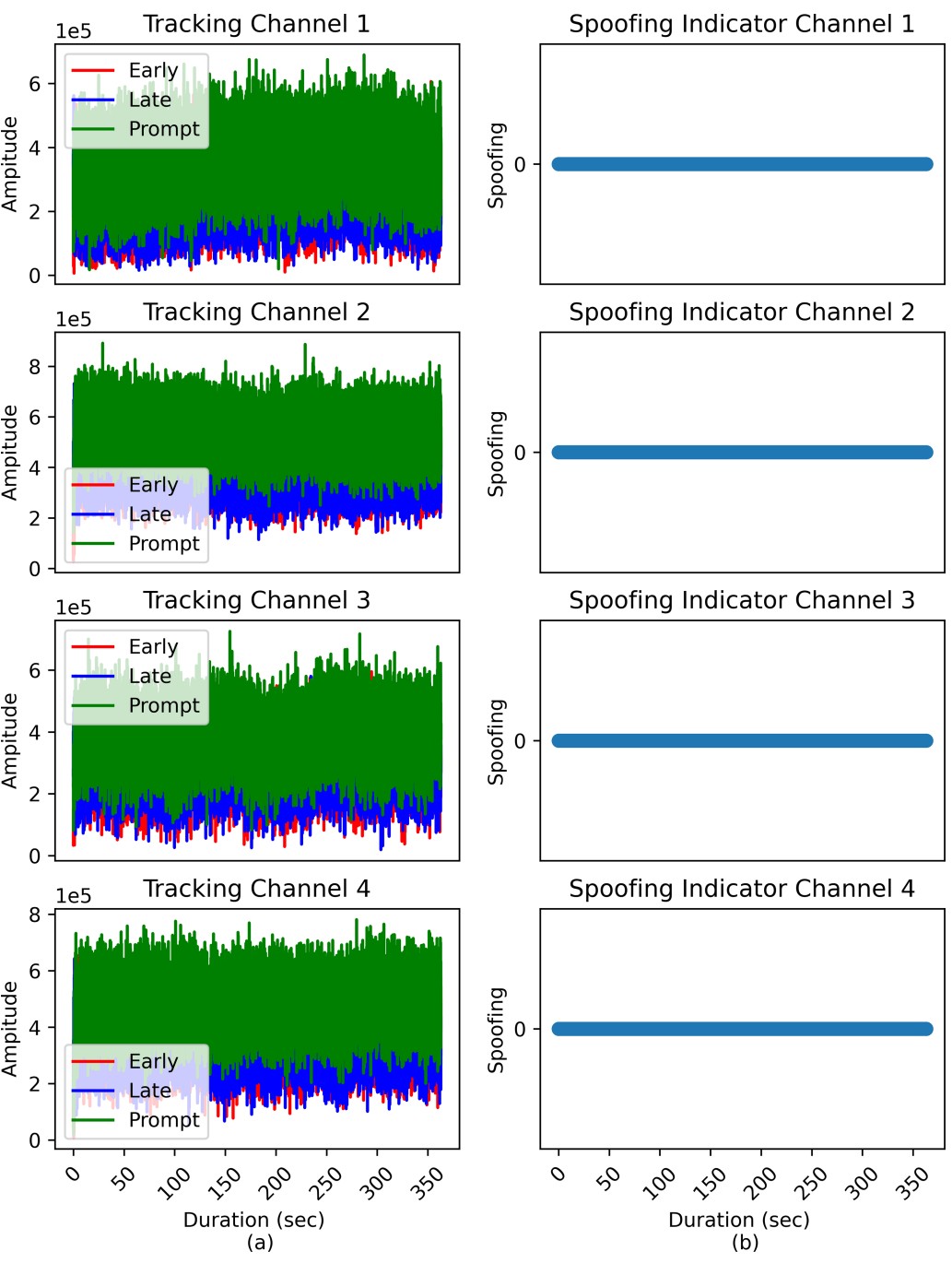

**Figure 8** Logistic regression base prediction of spoofing attacks in clean static data set.

model predicted no spoofing at all epochs present in the data set, which is true according to the given scenario. Next, the ds8 data set was fed to the prediction module of the model, and the results obtained can be observed in Fig. 9. During this case, the model predicted no

**Table 4** Inference time calculated for each machine learning classification model on an Intel Core i5 (10th gen 2.5 GHz) processor with 8 GB RAM.

| Model | Inference time (mean ± std. dev. of 3 runs, 10 loop each) | Time efficiency (dB) |
|---|---|---|
| KNN | 2.02 s ± 248 ms per loop | 24.53 |
| Logistic regression | 1.76 ms ± 44 μs per loop | 165.45 |
| Decision tree classifier | 3.45 ms ± 232 μs per loop | 151.98 |
| Support vector classifier | 6.89 s ± 84.7 ms per loop | 0 |

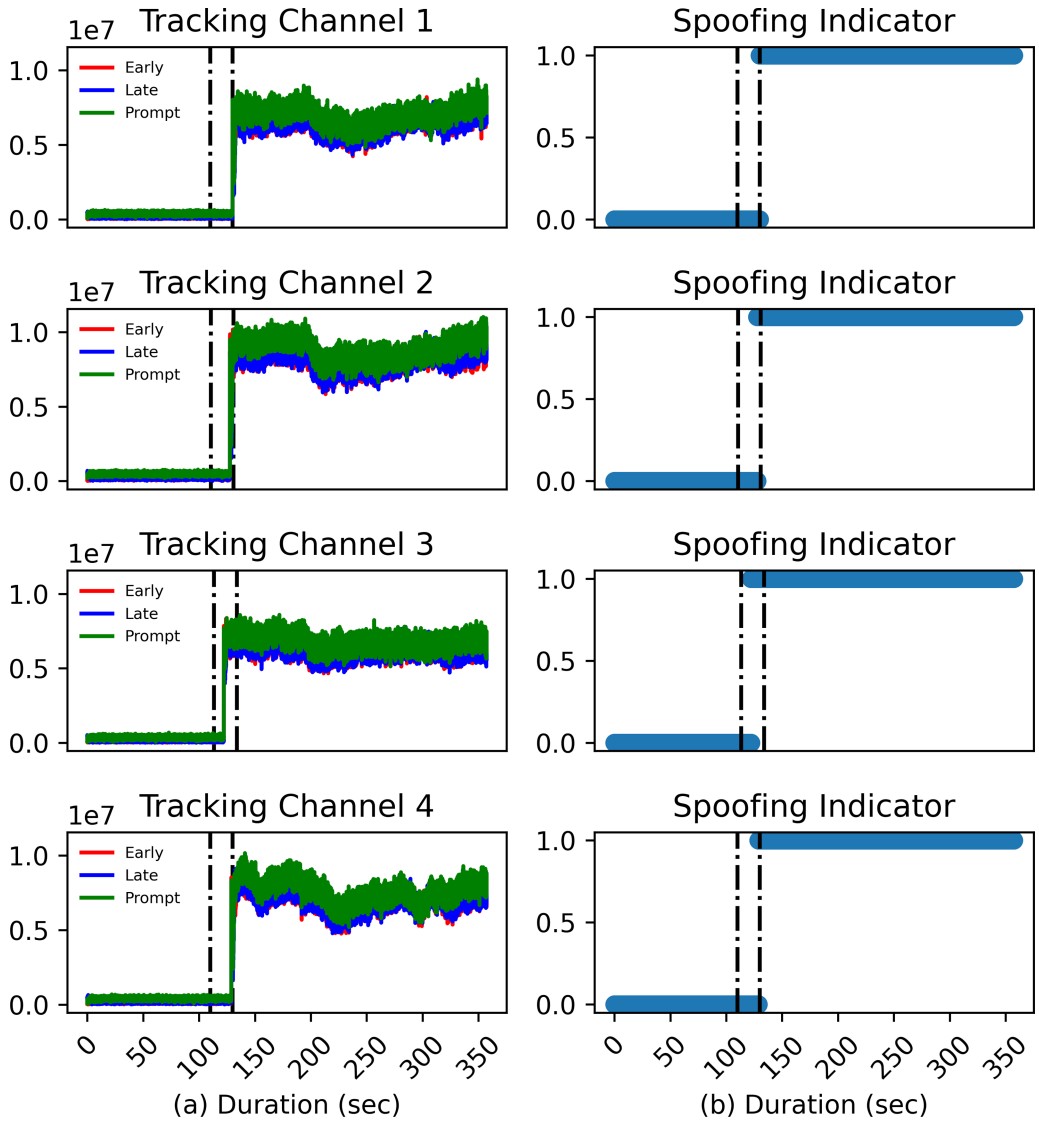

**Figure 9** Logistic regression base prediction of spoofing attacks in ds8 data set.

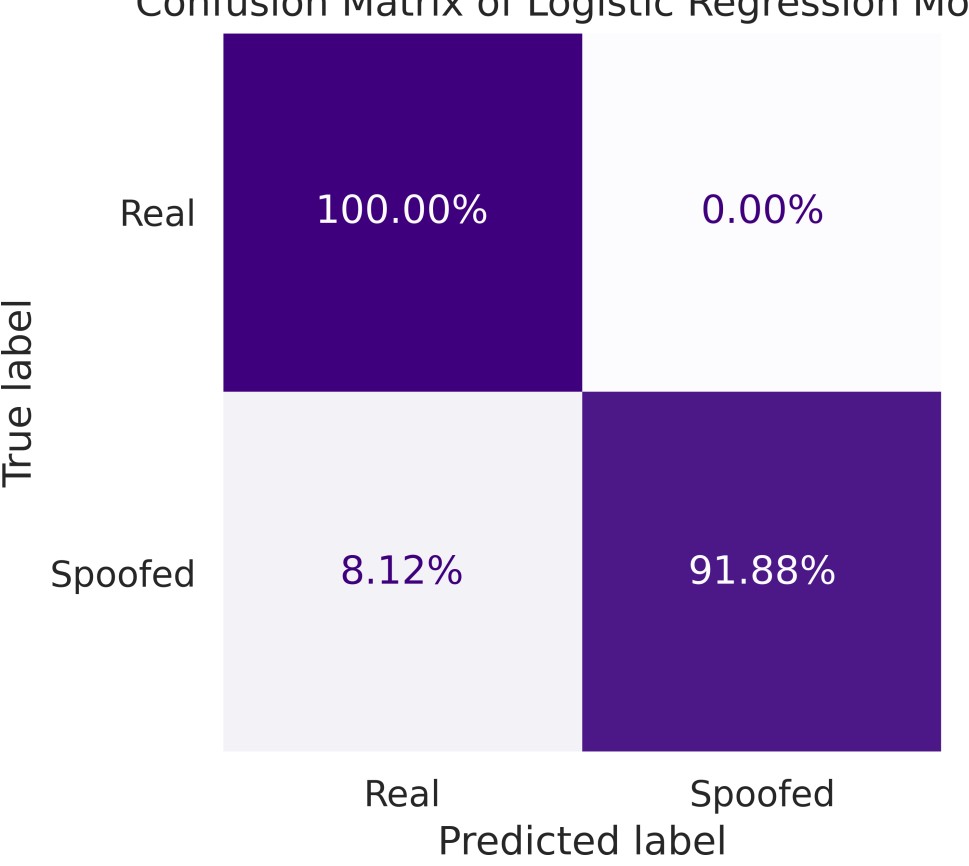

**Figure 10** Confusion matrix display between the actual target variable and predicted target variable by logistic regression model.

spoofing from epochs 1–110 (sec), and it jumped from 0 to 1 exactly where the spoofing scenario starts.

A two-dimensional confusion matrix is employed for performance evaluation of the model. Fig. 10, depicts the confusion matrix where cell (1,1) corresponds to the TP values, cell (1,2) represents FP values, cell (2,1) represents TN, and cell (2,2) corresponds to FN values. It can be observed that the model predicted a non-spoofing scenario with 100% accuracy for the ds8 data set and 91.88% of the time it accurately predicted a spoofing scenario with a FN rate of 8.12% while providing a significantly improved computational cost.

## CONCLUSION

This paper addresses detection of structured interference GNSS signals, in particular SCER attack, by leveraging machine learning techniques. To observe the efficacy of distinguishing between authentic and counterfeit signals, a comprehensive comparative analysis is undertaken between four ML classification models: logistic regression, SVM, KNN and decision tree. These models are trained and tested using the early, prompt and

late power of DLL correlators. Real and spoofing signals are analyzed for variation of $C/N_0$ values and the correlation power levels to justify the choice of decision parameters. It is seen that the $C/N_0$ of spoofed signal is carefully managed and is identical to the real signal except for a small drop at epochs where the tracking channels are taken under control of the spoofer. On the other hand, the DLL correlator power levels exhibit an abrupt change at the instant the spoofing signal is injected exhibiting significant merit to be considered as decision parameter. The trained models are validated against several performance metrics, including accuracy, precision, recall, F1-score, and area under the ROC curve. SVM and logistic regression classifiers exhibit superior performance having F1-score of 94%. However, computational efficiency of logistic regression supersedes the SVM, providing a huge gain of 165$dB$. For detailed analysis, confusion matrix for logistic regression is computed. It is seen that logistic regression model can predict non-spoofing instances with 100% accuracy. Similarly, for spoofing instances, the model performs 91.88% accurate predictions with a 8.12% FN rate. Therefore, based on superior performance metrics, logistic regression is chosen as the classifier for detection of SCER spoofing attack.

### Funding
This research has received funding from King Saud University through Researchers Supporting Project number (RSP2024R387), King Saud University, Riyadh, Saudi Arabia. The funder had a role in study design by structuring the approach the study took; data collection and analysis by determining which mathematical software and approaches could be used; decision to publish by providing permission to publish this study; and preparation of the manuscript by reviewing the content and providing feedback.

### Grant Disclosures
The following grant information was disclosed by the authors:
King Saud University through Researchers Supporting Project number: RSP2024R387.

### Competing Interests
The authors declare there are no competing interests.

### Author Contributions
- Imtiaz Nabi conceived and designed the experiments, performed the experiments, analyzed the data, performed the computation work, prepared figures and/or tables, authored or reviewed drafts of the article, and approved the final draft.
- Salma Zainab Farooq conceived and designed the experiments, performed the experiments, analyzed the data, performed the computation work, prepared figures and/or tables, and approved the final draft.
- Sunnyaha Saeed conceived and designed the experiments, performed the experiments, analyzed the data, performed the computation work, prepared figures and/or tables, and approved the final draft.

- Syed Ali Irtaza conceived and designed the experiments, performed the experiments, analyzed the data, prepared figures and/or tables, authored or reviewed drafts of the article, and approved the final draft.
- Khurram Shehzad conceived and designed the experiments, performed the experiments, prepared figures and/or tables, and approved the final draft.
- Mohammad Arif conceived and designed the experiments, performed the experiments, analyzed the data, prepared figures and/or tables, authored or reviewed drafts of the article, and approved the final draft.
- Inayat Khan conceived and designed the experiments, prepared figures and/or tables, and approved the final draft.
- Shafiq Ahmad conceived and designed the experiments, prepared figures and/or tables, authored or reviewed drafts of the article, and approved the final draft.

## Data Availability

The data and code are available at figshare: Arif, Mohammad (2024). dtb (1).xlsx. figshare. Dataset. https://doi.org/10.6084/m9.figshare.25744212.v1.

The original dataset Texas Spoofing Test Battery (TEXBAT) is available at: https://radionavlab.ae.utexas.edu/texbat.

## Supplemental Information

Supplemental information for this article can be found online at http://dx.doi.org/10.7717/peerj-cs.2399#supplemental-information.

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
