# Peer review of "Leveraging machine learning for the detection of structured interference in Global Navigation Satellite Systems"

_PeerJ Computer Science, doi:10.7717/peerj-cs.2399_

## Round 0.1 · original submission · Major Revisions

I have received reviews of your manuscript from scholars who are experts on the cited topic. They find the topic interesting; however, several concerns must be addressed regarding contributions, the proposed methodology, and experimental results. These issues require a major revision. Please refer to the reviewers’ comments at the end of this letter; you will see that they advise you to revise your manuscript. If you are prepared to undertake the work required, I would be pleased to reconsider my decision. Please submit a list of changes or a rebuttal against each point being raised when you submit your revised manuscript.

Thank you for considering PeerJ Computer Science for the publication of your research.

With kind regards,

Reviewer 1 ·

Basic reporting

No Comment

Experimental design

No Comment

Validity of the findings

No Comment

Additional comments

The article is entitled as "Leveraging Machine Learning for the
Detection of Structured Interference in
Global Navigation Satellite Systems". This study explores the
application of machine learning techniques for discerning authentic signals from counterfeit ones. The
proposed framework leverages data from six delay lock loop (DLL) correlators as intrinsic features to
train four distinct machine learning models: Logistic Regression, Support Vector, K Neighbors, and
Decision Tree classifiers. The accuracy of these models has been compared to select the most suitable
one for detecting spoofing instances with minimum computational complexity. Although the paper is well designed, there are major concerns; Revise it according to the suggestions given below
1. The abbreviations should be explained when it appears for the first time. In the abstract please specify the outcome of the results. It will show how the proposed model is better than the existing methods
2. Contribution of the work should be rewritten in a meaningful way.
3. Please clarify the research objectives and ensure they are clearly defined and unambiguous.
4. What are the major functions of the devised model in this field provide with its advantage.
5. Authors need to include a table summarizing the main notation that has been used in the paper, which currently is quite excessive,
6. Explain more detail about the GNSS Spoofing.
7. The authors should provide more information on the experimental setup used to evaluate the proposed model. The dataset description must be in more elaborately.
8. Results are well presented with more comparison tables.
9. Authors need to add more technical details to justify how the proposed approach is different from other existing approaches.
10. In my opinion, it is more important to think of the applicability aspect of the technology for practical users.
11. A thorough proofreading/restructuring/grammar/sentence formation and spelling checking of this article is essential.

Reviewer 2 ·

Basic reporting

The paper does not explicitly assess the impact and novelty, but the approach of using machine learning for GNSS spoofing detection is novel and contributes to the field.
Some suggestions to improve clarity and consistency:
Abstract
Specify what “GNSS-based applications” mean, giving examples for better context.
Reorganize sentences to improve logical flow so that each idea flows smoothly into the next.
Shorten sentences where possible to maintain brevity without losing key information.
Focusing on computational aspects, clarify specific advantages of logistic regression over the support vector classifier.

The introduction effectively sets the stage for the paper by discussing the importance and challenges of GNSS spoofing. Here are some suggestions to improve clarity and consistency:
Clearly state the main purpose of the research early in the introduction.
Provide more details on why current spoofing detection techniques fail against advanced attacks.
Emphasis the importance of using machine learning for this research.
Literature review
Summarize key points more succinctly to avoid overwhelming the reader with too much detail.
Highlight gaps in existing research more clearly to justify the need for the current study.
Use consistent terminology when discussing GNSS spoofing and related technologies.
Organize the review logically, moving from historical context to current challenges and then to proposed solutions.

Experimental design

The methodology is well-structured and detailed, allowing for reproducibility. However, the rationale for choosing the four specific machine learning models (Logistic Regression, Support Vector, KNeighbors, and Decision Tree classifiers) should be elaborated. It would be helpful to explain why other models, such as neural networks, were not considered.
Provide more details about why specific machine learning techniques were chosen for this study.
Highlight the unique contributions of this research compared to existing studies.
Summarize the key features and advantages of each machine learning model to justify their selection.
Clearly outline the data collection and feature extraction process.
Provide more details about the data preprocessing and feature extraction steps.

Validity of the findings

Discussion: The discussion section of the paper is incomplete because the presented method is not compared with other existing methods and does not focus on the shortcomings of the synchronization of the device generating the spurious signal with the real signal. In this section, the proposed method should be compared with other similar studies and issues such as the effects of potential demodulation errors should be addressed. In this way, the results of the paper can be evaluated more comprehensively and the effectiveness of the proposed method can be better understood.

Additional comments

The most important part of the article is the discussion section, this section is not enough. A comprehensive discussion supported by references is necessary.

---

## Round 0.2 · Minor Revisions

Most concerns raised by the reviewers have been addressed satisfactorily; however, the paper still needs further work regarding the discussion and conclusion sections. The conclusion section is just summarizing the work. The acronyms for the performance metrics should be defined before using them (see Section 5.2). Please revise all acronyms throughout the manuscript. Several variables used in all equations are not defined. These issues require a minor revision. Please submit a list of changes or a rebuttal against each point that is being raised when you submit your revised manuscript.

Thank you for considering PeerJ Computer Science for the publication of your research.

With kind regards,

---

## Round 0.3 · accepted · Accept

All concerns raised by the reviewers have been satisfactorily addressed, therefore, I am pleased to inform you that your work has now been accepted for publication in PeerJ Computer Science.

Thank you for submitting your work to this journal. On behalf of the Editors of PeerJ Computer Science, I look forward to your continued contributions.

With kind regards,